# Performance Prediction of Durum Wheat Genotypes in Response to Drought and Heat in Climate Change Conditions

**DOI:** 10.3390/genes13030488

**Published:** 2022-03-10

**Authors:** Marco Dettori, Carla Cesaraccio, Pierpaolo Duce, Valentina Mereu

**Affiliations:** 1Agricultural Research Agency of Sardinia, Viale Trieste 111, 09123 Cagliari, Italy; vmereu@agrisricerca.it; 2Institute of BioEconomy (IBE), National Research Council (CNR), Traversa La Crucca 3, 07100 Sassari, Italy; carla.cesaraccio@ibe.cnr.it (C.C.); pierpaolo.duce@ibe.cnr.it (P.D.); 3Impacts on Agriculture, Forestry and Ecosystem Services (IAFES) Division, Euro-Mediterranean Center on Climate Changes (CMCC), Via E. de Nicola 9, 07100 Sassari, Italy

**Keywords:** climate change, drought tolerance, abiotic stress, crop modelling, durum wheat

## Abstract

With an approach combining crop modelling and biotechnology to assess the performance of three durum wheat cultivars (Creso, Duilio, Simeto) in a climate change context, weather and agronomic datasets over the period 1973–2004 from two sites, Benatzu and Ussana (Southern Sardinia, Itay), were used and the model responses were interpreted considering the role of DREB genes in the genotype performance with a focus on drought conditions. The CERES-Wheat crop model was calibrated and validated for grain yield, earliness and kernel weight. Forty-eight synthetic scenarios were used: 6 scenarios with increasing maximum air temperature; 6 scenarios with decreasing rainfall; 36 scenarios combining increasing temperature and decreasing rainfall. The simulated effects on yields, anthesis and kernel weights resulted in yield reduction, increasing kernel weight, and shortened growth duration in both sites. Creso (late cultivar) was the most sensitive to simulated climate conditions. Simeto and Duilio (early cultivars) showed lower simulated yield reductions and a larger anticipation of anthesis date. Observed data showed the same responses for the three cultivars in both sites. The CERES-Wheat model proved to be effective in representing reality and can be used in crop breeding programs with a molecular approach aiming at developing molecular markers for the resistance to drought stress.

## 1. Introduction

### 1.1. Climate Change: Overall Projected Effects

Climate change in the twenty-first century is projected to cause increasing mean air temperatures, more frequent and intense extreme events such as longer lasting heatwaves and droughts, and more variable precipitation and surface water flows, unless strong mitigation actions occur in the next decades [1]. IPCC reports increasing air temperatures ranging from 1.0–1.8 °C under SSP1-1.9, 2.1–3.5 °C under SSP2-4.5, and 3.3–5.7 °C under SSP5-8.5) where, “SSPx refers to the Shared Socio-economic Pathway describing the socio-economic trends underlying the scenario, and y refers to the approximate level of radiative forcing (in W m^−2^) resulting from the scenario in the year 2100. SSP1-1.9 represents the low end of future emissions pathways. At the opposite end of the range, SSP5-8.5 represents the very high warming end of future emissions pathways from the literature. SSP2-4.5 represents a scenario with stronger climate change mitigation and lower GHG emissions” [1]. Climate change has already affected food security in several regions, with negative impacts especially at lower latitudes, while at high latitudes positive impacts have been recorded for some crops [2]. Recently released global projections of crop yields show an emergence of climate impacts (before 2040) on the major breadbasket regions, with larger losses for maize, soybean and rice and additional gains for wheat [3]. The projected positive effect for wheat is mainly due to the stronger CO_2_ response of C3 crops with respect to C4 crops and the expected increase in wheat yields at high latitudes that are currently limited by non-optimal minimum air temperatures [3]. Notwithstanding the great uncertainty in the scientific debate about the effects of CO_2_ on crop yields [4,5], due to the physiological and genetic complexity of the phenomena [6], the ability of crop models to capture the main effects of CO_2_ on crop yields under various growing conditions is fully acknowledged. However, high uncertainty still remains in crop model responses under high CO_2_ concentrations and further investigations are needed [6]. As an example, high temperatures may lower the beneficial effects of increased CO_2_ by reducing grain number, size and quality, as shown in rice [7] and in soybean [8,9]. In addition, increased levels of CO_2_ have negative impacts on food quality, by reducing the content of micronutrients such as iron and zinc [10], and on the protein content of cereals [11,12], with detrimental effects on the baking quality of wheat [13]. Importantly, the net effect of elevated CO_2_, lower growing-season rainfall and high temperature will likely increase ‘haying-off’, thereby limiting production of rain-fed wheat in Mediterranean-type environments [14]. As a result, instability in yields owing to increasing temperatures and higher frequency of extreme events may overcome the positive effects of a slight temperature increase [15]. Focusing on cereals, the negative impact of climate change on yields is very likely due to heat stress, increased plant water demand causing a higher transpiration rate, and a shortened growing period as well as anticipated maturity [16,17,18,19,20,21,22].

Observations show decreases of wheat and maize yields due to climate change in many low-latitude areas, while increases are reported for high-latitudes during the recent decades [2]. Global yield projections show yield decreases for maize ranging from −6% under SSP1-2.6 to −24% under SSP5-8.5, while for wheat the global projected changes in crop yield range from +9% under SSP1-2.6 to +18% under SSP5-8.5, and for rice from +3% under SSP1-2.6 to +2% under SSP5-8.5 by the end of the century [3]. The larger yield losses are expected at lower latitudes, while at higher latitudes potential yield gains are projected, even if with high uncertainty associated to simulations with the most pessimistic scenario (SSP5-8.5) [3]. In addition, maize yields are expected to be highly affected by climate change throughout Europe, while wheat yields could even increase as a consequence of more favorable conditions projected for Northern Europe [23].

However, adaptation strategies such as cultivar choice focused on drought tolerant genotypes, changes in planting dates and/or in irrigation scheduling, may counterbalance or even outweigh the effects of climate change [22,24,25]. From this point of view, using conventional breeding as well as a molecular approach focusing on the relationship between water stress tolerance and expression of specific genes will greatly help to develop better adapted crops for projected harsh growing conditions. In this context, plant genomic research is crucial to provide information related to the possible mechanisms involved in abiotic stress tolerance where an increasing number of genes, transcripts and proteins are involved in stress response pathways [26]. Likewise, increased water-use efficiency as well as soil conservative management techniques will become crucial goals in the next decades [27].

Concerning climate projections in the Mediterranean Region, in the next decades the effect of climate change on agriculture will very likely result in increasing plant water stress, decreasing crop yields, especially in spring sown crops, and increasing yield variability basically due to abiotic stresses such as heatwaves and droughts [16,28]. The Mediterranean Region is considered a “hot-spot”, with observed rates of climate change exceeding the global trends for most variables and future projections showing a temperature increase higher than 20% of the global average and decreases in precipitation especially for central and southern areas [29,30]. Consequently, in this area the impacts are likely to exceed the global average trend [29,31]. However, these impacts will be likely related to crop and cultivar characteristics, including their genetic mechanisms of water stress tolerance as well as their response to CO_2_ in terms of increase of biomass and water use efficiency. Moreover, other issues linked with climate change such as limitations in available lands, soil erosion, salinization, decreasing natural rainfall and increasing population may exacerbate the predicted negative impact of global warming, especially on the southern side of the Mediterranean Region [32]. For example, Maghreb countries as well as Egypt and Libya, are bound to face water scarcity due to the average annual growth of population and the reduction of long-term freshwater resources [33]. Therefore, food safety and security in the Mediterranean Region are expected to be seriously threatened as a result of expected climate and socio-economic changes [29].

### 1.2. Projected Effects on Durum Wheat Production

In the Mediterranean Basin durum wheat (*Triticum turgidum* L. subs. *durum* [Desf.]) is the most widely grown crop, accounting for half of the total world production [34]. Italy is the main producer of the area, followed by Turkey, Southern France, Algeria, Morocco, Syria, Greece, Spain and Tunisia.

Concerning the effects of climate change on durum wheat production (i.e., yield and grain quality), warmer and drier climate is projected to increase the risk of yield losses, especially for temperature increases exceeding 2 °C across the whole Mediterranean area [35]. Even greater yield reductions with a decrease of 30–50% as a result of a 4 °C increase in temperature were projected in the province of Foggia (Apulia, Southern Italy) [22]. In another study [36], an increase in grain yield of about 10% in the “Anomaly_2” scenario (+1.7 °C; 10.4% rainfall mean reduction) in response to elevated CO_2_ together with a decrease of about 8% under “Anomaly_5” scenario (+4.2 °C; 20.9% rainfall mean reduction) in the outstanding durum growing area of Capitanata (Apulia, Italy) were predicted. In a more recent study [37], negative impacts (−30% by 2100 under the business-as-usual scenario, RCP (Representative Concentration Pathway) 8.5, no CO_2_ effect) in central and southern Italy (e.g., Apulia, Basilicata, Campania, Lazio, Sardinia and Sicily) were reported, together with increases in northern areas (e.g., Po river plains), especially if the CO_2_ effect is included in the simulations.

As for grain quality, durum wheat is basically used for pasta (Italy), cous-cous (North Africa) and bread (semolina high volume heart breads and flat breads). These end-products are traditionally associated with the Mediterranean cuisine as the basis of Mediterranean diet. Hence, grain quality is crucial to meet the requirements of downstream, large-scaled processing activities such as milling and pasta-making. Grain size in barley and wheat is negatively affected by high temperatures during grain filling [38,39,40,41]. Hence, climate change might cause an increase of shrivelled grains with low yield in semolina, thereby jeopardizing the technological value of durum wheat production. Therefore, the negative effect of increasing temperatures and drought on grain production, including grain quality and yield components, is an issue of utmost importance not only for agriculture but also for the food industry of the entire Mediterranean Region. Thus, a negative impact on durum wheat caused by climate change may also result in dramatic socio-economic consequences.

### 1.3. Crop Modelling

Given this context, using crop models to analyze the responses of crops across different environments in order to assess the impact of climate change becomes crucial [42,43] both to study the plant responses and to find adaptation and mitigation strategies in association with the study of gene-induced stress tolerance. The combined contribution of crop modelling and genetics can play an innovative and crucial role in targeting cultivar choice in different environments and outweighing risks. To date, only a relatively low number of studies based on crop modelling focused on crops under Mediterranean conditions have been carried out [16,35,36,37,44,45,46,47,48,49,50,51].

Among several crop simulation models developed since the sixties [52], the Decision Support System for Agrotechnology Transfer (DSSAT) [53] has been extensively used worldwide over the last 20 years for most major crops and many different applications including: (i) simulated management options such as fertilization, irrigation, pest management; (ii) site-specific farming and study of potential impacts of climate change on agricultural production (see [36] for an exhaustive bibliography). The basis for the DSSAT cropping system model design is a modular structure of which CERES-Wheat is a component model continuously refined and modified over the years [54,55,56]. CERES-Wheat simulates crop growth, development and yield taking into account the effects of weather, management, genetics, soil water, carbon (C) and nitrogen (N_2_). This model provided successful performance under a wide range of agro-climatic conditions [57]. In order to evaluate the performance of the model [58], using various statistical tests to analyze simulated and observed results from several studies across Asia and Australia, proved the good ability of this model to predict phenology (i.e., anthesis and maturity) and, to a slightly lesser extent, grain and biomass yields. Furthermore, they underlined good performance in favorable conditions (e.g., high-yielding and/or irrigated environments). In contrast, results were less satisfactory in low-yielding environments. Importantly, these authors emphasized the crucial role of long-term data-sets to better evaluate the effectiveness of this model in representing reality.

In the Mediterranean environments, the CERES-Wheat model has been used in a limited number of studies [36,37,45,50,51,59,60,61,62]. In particular, refs. [45,50], focusing on the use of long-term data-sets (approximately 30 years) and statistical indices to assess the model performance, found that the CERES-Wheat model provided good to fair predictions of production (i.e., grain yield), with a tendency to overestimate, and good to very good predictions of phenology (i.e., anthesis date). In contrast, predictions of grain quality (i.e., grain weight and grain number) proved to be reliable but less satisfactory.

This study applies the CERES-Wheat crop model included in DSSAT v. 4.0 to assess the adaptation of cultivar choice as well as the environmental effect of genetic mechanisms involved in water stress conditions with a special focus on Dehydration Responsive Element-Binding (*DREB*) related genes in durum wheat, in order to evaluate the simulated negative impact of increasing temperatures and decreasing rainfall on grain production and phenology of three durum wheat genotypes grown in two different Mediterranean environments in Southern Sardinia (Italy).

## 2. Materials and Methods

### 2.1. Experimental Setting

This study was carried out at Benatzu and Ussana, two sites from AGRIS (Agricultural Research Agency of Sardinia) experimental farm “S. Michele” (Lat. 39°24′ N, Long. 9°5′ E; about 20 km S from the sea) in the Campidano plain, the main durum growing area of Southern Sardinia, Italy. The climate is Mediterranean, with warm and dry summers and mild winters. Mean air temperatures range from 4.8 °C in January and 33.0 °C in August. Precipitations are concentrated in autumn, winter and early spring, with a long-term annual amount of about 450 mm. The area shows a great soil variability due to its ancient geological origin. Ussana soil (114 m a.s.l.) is a *Petrocalcic Palexeralf* [63]. It is a sandy clay loam soil, with a percentage of sand greater than 50%, characterized by alluvial conglomerate substrate, in a weak red colored clay matrix. The drainage is moderate and the stone percentage is about 20%. This soil is located in a hilly area and accounts for medium- and low-fertility durum growing areas of Sardinia and the Mediterranean. Benatzu soil (80 m a.s.l.), is a *Vertic Epiaquet* [63]. It is a clay loam soil with a soil substrate alluvial gravel, a fraction of stones of about 30% and a clay percentage of about 40%. This soil is located in a flat area and accounts for the most fertile durum growing areas of Sardinia and the Mediterranean.

### 2.2. Experimental Data

Daily maximum and minimum air temperatures (°C) and annual rainfall totals (mm) over the period 1973 to 2004 were recorded by an automatic weather station (SILIDATA AD2, SILIMET s.r.l., Modena, Italy) located in the experimental farm. Daily global solar radiation values (MJ m^2^ d^−1^) were estimated using the software RadEst3.00, where: (i) radiation is calculated as the product of the atmospheric transmissivity of radiation times the radiation outside the earth atmosphere and (ii) the atmospheric transmissivity of global solar radiation is estimated based on the difference between maximum and minimum temperatures [64]. Several physical and chemical characteristics of soils required as inputs by the model were determined at both sites (Table 1): content in sand (%), silt (%), clay (%), total N (%), pH in H_2_O, C.E.C. (Cation Exchange Capacity in cmol kg^−1^), organic C (%), organic matter (%), texture, color, runoff value, slope and a fertility factor. The analysis procedures are described in DM 13.09.1999, points: II.4 and II.5 for Sand, Silt, Clay and Texture; VII.1 for Organic carbon and Organic matter; III.1 for pH in H_2_0; XIV.1 for Total nitrogen; and XIII.2 for C.E.C.

Management and durum wheat performance data over the period 1973 to 2004 from both sites were taken from the evaluation trials of the Italian Durum Wheat Network (http://qce.entecra.it/RISULTATI.htm, (accessed on 1 December 2021)). The experimental design consisted in a triple lattice with 8-rowed plots. Each plot was 5.9 m long and 1.5 m wide with an approximate surface of 10 m^2^ and rows spaced 0.18 m apart. Plant density was about 350 viable seeds m^−2^.

### 2.3. Cultivar Description

Creso, Duilio and Simeto, three hallmark Italian durum wheat cultivars, well adapted to Mediterranean environmental conditions, were used to test the performances of the CERES-Wheat model. In addition to their agronomic and economic importance, these varieties were chosen for the availability of reliable long-term experimental data. Creso, released in 1973, is a medium-late, short variety with good grain quality. Despite its longstanding cultivation, it is still widespread in the high rainfall spring areas of Central Italy. Duilio, released in 1984, is an early-medium, medium-tall variety, well adapted to the durum growing areas of Southern Italy owing to its high-yielding potential, grain quality and resistance to drought. Simeto, released in 1988, is an early and short genotype with good performances both in yield and grain quality, especially in the dry areas of Southern Italy. This cultivar still ranks among the most widespread in Italy for the production of certified seed (https://www.crea.gov.it/web/difesa-e-certificazione/-/statistiche-di-certifcazione-superfici-controllate, (accessed on 1 December 2021)).

### 2.4. Molecular Responses to Drought Stress

These three cultivars had previously been studied in regards to their molecular responses to abiotis stresses in general and drought stress in particular. In this study, the expression of the endogenous *DREB2A*-homologous gene activation, belonging to the Dehydration-Responsive Element-Binding (*DREB*) transcription factor gene family, was considered through RT-PCR analyses obtained from time-course experiments of drought stress both in controlled greenhouse and in field conditions [65,66].

### 2.5. Model Simulations

The CERES-Wheat model, included in the Decision Support System for Agrotechnology Transfer (DSSAT) version 4.0 [53,67] was used to perform crop growth simulations of Creso, Duilio and Simeto. This model describes daily both phenology and growth in response to environmental factors (e.g., soil properties and weather patterns) and management. The model, which includes subroutines to simulate soil and crop water balance and nitrogen balance, can be used to simulate the effects of nitrogen deficiency and soil water deficit on photosynthesis and pathways of carbohydrate allocation in plants.

CERES-Wheat had already been calibrated and validated in the test area for these cultivars [45]. An iterative procedure for minimizing the differences between predicted and observed values to obtain the genetic coefficients values was used [68]. In particular, observed and predicted values of grain yield (kg ha^−1^), anthesis date (days after planting, dap) and average seed weight (g) were compared and the cultivar coefficients were modified until the model responses matched the real data or fell within a defined error threshold. Table 2 exhibits the genetic coefficients found for the three cultivars in the area.

Concerning the simulation runs, 1 August was set as the starting day for each year. The cropping season was between October and June of the following year. The planting date was set on the observed date of each year, depending on the amount of natural rainfall fallen from autumn until late early winter. The end of the growing season was set according to the observed harvest dates for each year. All agronomic information, such as previous crops and fertilizer management, was set in the experimental file. The following data were registered as initial conditions: previous crop, sowing depth and dates, row spacing, plant population, fertilizer applications and dates, harvest dates. The same data were subsequently set as inputs in the experimental simulation design.

The following indices based on simple and squared differences between predicted and measured values were calculated: normalized Root Square Error (nRMSE), index of agreement (D-index) and Coefficient of Residual Mass (CRM) [69]. Ideally, a model reproduces experimental data perfectly when nRMSE is 0 and D-index is 1 [58]. CRM measures the tendency of the model to over- (i.e., negative values) or under-estimate (i.e., positive values) observed data [70].

### 2.6. Meteorological Trends and Climate Scenarios

Mean annual air temperature data (°C), along with Standardized Anomaly Index (SAI) values, and the annual and seasonal amount of rainfall (mm) over the period 1974 to 2004 were observed in order to evaluate the real trend of air temperature and rainfall in the study area.

As for climate scenarios, a set of 48 synthetic climates based on global and regional climate model simulations predicting a substantial drying and warming over the Mediterranean Region by the end of the century, with annual precipitation decrease exceeding −25–30% and warming exceeding +4–5 °C compared to the actual climate, was developed [71,72]. The baseline air temperature, as well as the precipitation records for the actual climate recordings over the 1973–2004 period at Benatzu and Ussana sites, were adjusted by between +1 and +6 °C at 1 °C intervals, and by between −5% and −30% at 5% intervals, respectively (Table 3). For more details concerning the pro and cons of the incremental approach followed in this study and for an exhaustive review, see [50].

This set of 48 synthetic scenarios was used in conjunction with the CERES-Wheat crop model to determine the potential effects of increasing temperatures and decreasing rainfall on crop production (i.e., grain yield and grain size), and phenology (i.e., anthesis date) of the three durum wheat cultivars Creso, Duilio and Simeto by scenario and site.

### 2.7. Calibration, Validation and Evaluation of CERES-Wheat Model Performances

The whole study period for Benatzu and Ussana sites, covers a time span of 30 years (1974–2004). Datasets used for calibration were: 1996–2004, 1997–2004 and 2000–2004 for Creso, Duilio and Simeto, respectively. Datasets used for validation were: 1974–1995, 1985–1996 and 1989–1999 for Creso, Duilio and Simeto, respectively. The differences in time span both in calibration and validation depend on the availability of data from the cultivar evaluation trials of the Italian durum wheat network owing to the different year of release of each cultivar. Detailed information about calibration, validation and evaluation of the CERES-Wheat model in the two experimental sites of the study area can be found in [45].

## 3. Results

### 3.1. Meteorological Trends

Trends in mean annual temperature (T_mean_) (°C) along with Standardized Anomaly Index (SAI) values, and annual and seasonal amounts of rainfall (mm) observed in the study area over the period 1974 to 2004, are shown in Figure 1 and Figure 2, respectively. The T_mean_ linear trend shows an increase of 0.44 ± 0.59 °C per decade (Figure 1A). Lower than average temperatures are prevalently scattered over the left-hand side of the SAI graph (approximately from 1974 to 1985), i.e., in the first years of the study period, whereas in the following years higher than average temperatures become more frequent (Figure 1B). A non-significant negative trend for annual and seasonal rainfall amounts was observed except in autumn, which showed a non-significant increasing trend (Figure 2A,B). This seasonal downward trend is clearer in winter than in spring and summer.

### 3.2. Calibration, Validation and Evaluation of CERES-Wheat Model Performances

Concerning grain yield predictions, the CERES-Wheat model provided good to fair performances for all three cultivars. As for phenology, all results proved the effectiveness of the CERES-Wheat model in predicting anthesis dates for these experiments. On the contrary, the model performances proved to be less effective for estimating the average seed with a tendency of the CERES-Wheat model to underestimate predictions. However, considering the results as a whole, statistical indices show that this model proves to be an effective tool to represent reality. For further details and an exhaustive presentation and discussion about results and model performances, see [45].

### 3.3. Climate Change Scenarios: General Responses

In order to evaluate the impacts of climate change on durum wheat production and phenology, the general analysis was performed using data sets from the whole study period (1974–2004). The experimental conditions observed during calibration and validation were left unchanged. Hence, weather was the only factor of variation.

The responses of the CERES-Wheat model to 48 simulated scenarios (Table 3) at the two experimental sites “Benatzu” and “Ussana” for the annual values of grain yield, anthesis date and average seed weight and for three durum wheat cultivars were analyzed by comparing observed and simulated values. Figure 3 shows the observed mean grain yield data in comparison with the CERES-Wheat model responses to 2 (mildest and worst scenarios, respectively) of the 48 simulated climate change scenarios for Creso (time span: 1974–2004), Duilio, (time span: 1985–2004) and Simeto (time span: 1989–2004) at Benatzu and Ussana sites, respectively. The mildest simulated scenario shows a +1 °C increase in temperature and a 5% reduction in rainfall compared to the actual mean temperatures and total rainfall amount, respectively. The worst-case scenario shows a +6 °C increase in temperature and a 30% lower annual rainfall. The detrimental effect on simulated yield determined by increasing temperatures and decreasing rainfall for all cultivars and sites cannot be questioned.

### 3.4. Climate Change Scenarios: Cultivar Responses

To compare the simulated impact of increased temperatures and decreased rainfall on each cultivar, the analysis was limited to the years when field trials, were conducted simultaneously for all cultivars (i.e., years 1990–2004 for Benatzu and 1989–2004 for Ussana). Figure 4 and Figure 5 exhibit the percentage reductions in grain yield of all cultivars between the mean values observed at Benatzu and Ussana, respectively and simulation results from twelve climate change scenarios with increasing temperatures (from +1 °C to +6 °C) and decreasing rainfall (6 scenarios with a 5% reduction and 6 scenarios with a 30% reduction in annual rainfall). For each cultivar, the predicted negative impact on grain yields rises steadily from the least unfavorable scenarios to the most severe ones.

Comparing the different responses of the three cultivars to simulated scenarios at Benatzu site, Creso (medium-late cultivar) proved to be the most sensitive, with the greatest yield reduction especially when temperature increases were combined with strong decreasing rainfall. For this cultivar, the reduction in grain yield from mean observed values ranged from 2.4% (scenario T1_R5) to 14.9% (scenario T6_R5) for a 5% lower annual rainfall amount (Figure 4A), and from 19.9% (scenario T1_R30) to 29.2% (scenario T6_R30) for a 30% decrease in annual rainfall (Figure 4B).

The reduction in grain yield of Duilio (early cultivar) and Simeto (early cultivar) ranged from 2.7% and 1.7% (scenario T1_R5) to 9.1% and 8.6% (scenario T6_R5), respectively, for a 5% decrease in rainfall (Figure 4A). The reduction in grain yield of Duilio and Simeto was much higher using a 30% rainfall decrease scenario, ranging from 22.4% and 21.4% (scenario T1_R30) to 25.5% and 26.4% (scenario T6_R30), respectively (Figure 4B).

Creso was also confirmed to be the most sensitive cultivar at Ussana site over the period 1989 to 2004, with a grain yield reduction ranging from 4.2% (scenario T1_R5) to 15.3% (scenario T6_R5) for a 5% rainfall reduction (Figure 5A), and from 26.0% (scenario T1_R30) to 33.3% (scenario T5_R30) for a 30% rainfall decrease (Figure 5B).

Duilio showed grain yield declines ranging from 3.3% (scenario T1_R5) to 10.0% (scenario T6_R5) for a 5% rainfall reduction (Figure 5A), and from 25.6% (scenario T2_R30) to 29.3% (scenario T5_R30) for a 30% rainfall decrease (Figure 5B).

A similar trend was observed for Simeto with a decrease of grain yield ranging from 6.7% (scenario T1_R5) to 11.2% (scenario T6_R5) for a 5% rainfall reduction (Figure 5A), and from 27.8% (scenario T2_R30) to 30.1% (scenario T6_R30) for a 30% rainfall reduction (Figure 5B).

In summary, the overall simulated effect of climate change scenarios characterized by increasing temperature and decreasing rainfall is a gradual reduction in grain yield for all cultivars and sites. This effect increases from the mildest to the worst-case scenarios. Interestingly, the CERES-Wheat model predicted greater grain yield reductions at the low-yielding site of Ussana than in the fertile soil of Benatzu. In particular, the overall average grain yield reduction for the three cultivars in all scenarios was equal to 16.2% and 19.0% at Benatzu and Ussana, respectively. 

The overall simulated impact of increasing temperatures and decreasing rainfall on kernel weight showed an opposite trend. CERES-Wheat simulations showed that kernel weight tends to increase slightly and this response is greater when annual rainfall amount decreases by 5% (Figure 6A and Figure 7A). In addition, the slight increase in kernel weight is greater at Ussana and this confirms the trend in observed data (Figure 6 and Figure 7 for Benatzu and Ussana experimental sites, respectively). No remarkable trend from the analysis of the different responses of each variety emerges.

The effects of the 48 climate change scenarios on phenology of durum wheat were determined by comparing predicted and observed anthesis dates of each cultivar. In general, a shortening effect on cycle length of durum wheat was observed. This response probably depends on the modelling approach on phenology used by the CERES-Wheat crop model, which simulates crop development rate as a function of temperature only. Figure 8 illustrates the general shortening effect of climate change scenarios on the crop growing cycle at Benatzu (Figure 8A) and Ussana (Figure 8B) experimental sites.

Based on the greater overall shortening effect at Ussana (medium-low fertility soil) than at Benatzu (high-fertility soil), soil fertility seems to play a remarkable role in reducing the growth duration of durum wheat. In addition, the difference between simulated and observed values increases moving from scenario T1 (temperature increase: +1 °C) to scenario T6 (temperature increase: +6 °C). Examining this shortening effect on each variety, Creso showed a more limited reduction at both sites when compared to the early genotypes Duilio and Simeto.

Table 4 summarizes the simulated impacts of two climate change scenarios (T1_R5 and T6_R30) on grain production and phenology of durum wheat using the CERES-Wheat crop model on three cultivars and two experimental sites.

Creso shows the lowest observed yield potential (observed mean yield: 3877 kg ha^−1^) and the largest percentage reductions in grain yield (mean percentage reduction: 31.2%) under the worst-case (T6_R30) climate change scenario when compared to Simeto (observed mean yield: 4093 kg ha^−1^; percentage reduction under T6_R30 scenario: 28.3%) and Duilio (observed mean yield: 4165 kg ha^−1^; percentage reduction under T6_R30 scenario: 27.3%). In general, Duilio exhibits the smallest simulated grain yield reduction under climate change with a decrease ranging from 3.0% (scenario T1_R5) to 27.3% (scenario T6_R30) and proves to be the most resilient genotype to increasing unfavorable conditions. Furthermore, Ussana was the most vulnerable environment to climate change conditions with a general grain yield reduction of 4.7% and 30.8%, respectively for scenarios T1_R5 and T6_R30, when compared to Benatzu (2.3% and 27.0% for scenarios T1_R5 and T6_R30, respectively).

As for grain size, the largest effects of the two climate change scenarios T1_R5 and T6_R30 on kernel weight were registered for Creso, with a grain weight increase ranging from 2.5% at Benatzu to 6.0% at Ussana. Moreover, at Ussana the simulated percentage kernel weight increase ranged from 2.1% to 4.2% for scenarios T1_R5 and T6_R30, respectively, and was higher than Benatzu (1.9 for the two scenarios, respectively).

The analysis of the differences between observed and simulated anthesis dates under climate change scenarios indicates a general anticipation of anthesis, with some differences among genotypes. In particular, the late genotype Creso shows a general reduction to increasing temperature scenarios, ranging from 3 days at both experimental sites (scenario T1_R5) to 8 and 10 days at Benatzu and Ussana, respectively, for scenario T6_R30. The responses of the early genotypes Simeto and Duilio indicate a slightly larger shortening effect, ranging from 2 and 3 days for scenario T1_R5 to 11 and 13 days at Benatzu and Ussana sites respectively, for scenario T6_T30.

### 3.5. Molecular Responses to Drought Stress

Previous RT-PCR experiments carried out using RNA extracts from different durum wheat cultivars, including Creso, Duilio and Simeto, showed an intense band at 500 bp, instead of the expected 450 bp, and two faint bands at 450 bp and 580 bp, respectively (Figure 9) [73].

Moreover, sequencing and aligning with TC85717 500 bp bands from all these cultivars, the presence of a short insert 53-bp was detected, revealing a complete homology with transcripts found in barley, homologous to *DREB2* genes and related to drought. The transcript isolated in barley derives from an alternative splicing of a gene, named *HvDRF1*, where Hv stands for *Hordeum vulgare*, generating three transcripts. The primer pair used for these experiments was compatible with these transcripts and produced fragments at about 580 bp, 500 bp and 450 bp, which is the same pattern observed in durum wheat. This result led to the conclusion that in durum wheat a homologous gene to *HvDRF1* is present and it was named Triticum durum Dehydration-Responsive Factor 1 (*TdDRF1*). Further studies revealed that this gene produces three transcripts by alternative splicing: *TdDRF1.1*, consisting of four exons, from E1 to E4; *TdDRF1.2*, consisting of three exons E1, E2 and E4; and *TdDRF1.3*, consisting of two exons E1 and E4 [74]. This gene and its three isoforms play a crucial role in conditioning and modulating the responses of cultivars to drought. In all genotypes, the *TdDRF1.2* transcript was always expressed at higher levels, the *TdDRF1.1* transcript was the least expressed and the TdDRF1.3 transcript was intermediate between TdDRF1.2 and *TdDRF1.1* transcripts. These results suggest a correlation between water stress and the expression profile of the *TdDRF1* gene and its transcripts.

## 4. Discussion

### 4.1. Meteorological Trends

The analysis of the historical weather data set covering the study area over the period 1974–2004 confirmed an overall trend with increasing temperatures and decreasing and/or more erratic precipitations. Mean temperatures (Figure 1A) showed an increasing rate in agreement with the observed trend in Europe during the last three decades [75,76]. In addition, our results confirm a negative yearly rainfall trend in the Mediterranean area [77]. The different trends shown in Figure 1A and Figure 2B have some relevant agricultural implications: (i) increasing autumn rainfall in rain-fed durum growing areas leads to a greater water storage in the soil but may also delay sowing, especially when associated with increasing mean precipitation [78]; (ii) decreasing rainfall in winter reduces soil moisture with detrimental effects on water uptake especially if combined with limited root growth due to increased temperatures and delayed sowing; (iii) increased temperatures or heat shocks in late spring may abruptly interrupt translocation of photosynthates during grain filling thereby exposing caryopses to the risk of ‘haying-off’ [14,79,80]. All these points may dramatically result in increased vulnerability particularly in the agricultural systems of the Mediterranean Region [81,82].

### 4.2. Calibration, Validation and Evaluation of CERES-Wheat Model Performances

Calibration and validation of the CERES-Wheat crop model in the study area was already discussed in [45], where full details are available. Importantly, this study emphasizes the crucial importance of using data from long-term experiments [58] to overcome poor performance of the model due to deficiencies in model inputs and experimental observations as well as to allow a proper calibration. In general, the values of the genetic coefficients determined in this current study are similar to those obtained by the few other ones conducted on durum wheat [60,61], with the exception of parameters G1, G2 and G3. Moreover, the good and excellent results of the model in predicting grain yield and phenology, respectively, confirm the observations of [58] in their review of the performance of CERES-Rice and CERES-Wheat models in rice-wheat systems of South Asia, China and Southeast Australia.

The model proved to be less satisfactory in the case of kernel weight. This was probably due to modelling inaccuracies in simulating underlying physiological processes under stressed and non-stressed conditions [83]. Interestingly, the combined overestimate of grain yield and the underestimate of kernel weight resulting in an overestimate of the number of kernel per unit area had already been remarked in previous studies [45]. Hence, these systematic errors might be due to either inconsistent estimation of the number of grains or differences between durum wheat and bread wheat. Of note, the analysis by site revealed a better performance of the model at Benatzu when compared to Ussana for both grain yield and kernel weight. This is likely due to the low fertility of Ussana soil making this site drought-prone and with a greater frequency of very low yields. In this context, the poorer performance of the CERES-Wheat model under low-yielding conditions was already known [58]. In addition, the tendency of CERES-Wheat to overestimate grain yield under water shortage conditions has already been underlined [84]. Finally, the model accuracy in predicting anthesis dates did not show any remarkable differences between sites.

### 4.3. Climate Change Scenarios: Cultivar and Molecular Responses

The negative effect of increasing temperatures and decreasing rainfall on simulated grain yields at the two experimental sites is clear (Figure 3). Interestingly, Ussana site, less fertile and negatively affected by rainfall decrease and water scarcity, showed the greater yield reductions when compared to Benatzu (Figure 4 and Figure 5). In summary, the yield reductions observed in our simulations are mostly consistent with other global [85,86,87,88,89] and regional studies [22,35,36], excluding the CO_2_ effect that was not taken into consideration in this study, limiting evaluation of the cultivar responses to climate stimuli.

Concerning grain quality, the slight positive effect of increasing temperature and decreasing rainfall on kernel weight (Figure 6 and Figure 7) is in contrast with findings on bread wheat showing a weight reduction due to high temperatures [39], heat shocks [90] and water stress [91] during grain filling. In all likelihood, these contrasting results may be associated with a lower correspondence between observed and predicted data for this trait during calibration and validation of the CERES-Wheat model [45].

As for phenology, the shortening of the growing period highlighted in this study is in agreement with other studies [19,37,92]. Interestingly, the reduction in growth duration from sowing to anthesis was larger for the early cultivars Duilio and Simeto when compared to the late cultivar Creso at both sites (Figure 8), confirming the adaptive role of earliness for durum wheat in drought prone environments [93]. Remarkably, the early genotypes Duilio and Simeto had a better yield performance than the late genotype Creso in both observed and predicted data (Table 4). Therefore, the CERES-Wheat crop model seems to capture fairly well the greater resilience shown by early genotypes in rain-fed Mediterranean conditions.

Regarding cultivar choice, this study confirms its potential key role as a farm-level adaptation measure to reduce the negative impacts of climate change on crop production [19]. In particular, an estimated avoidance of 10–15% yield reduction due to cropping adaptations such as changing cultivars and sowing times has been reported in the literature [15]. Thus far, a little effort has been made to understand the effect of cultivar choice and its role in tackling the detrimental effects of increasing temperatures and decreasing (or more erratic) rainfall on crop production. Our study shows a negative impact of harsh scenarios (i.e., increased air temperatures and decreased rainfall) on grain yield for all cultivars and sites. However, this detrimental effect can be mitigated by: (i) early sowing and (ii) replacing late genotypes with early ones. Concerning the latter point, our simulations show a percentage grain yield reduction from −31% (−29.2% and −33.1% at Ussana and Benatzu, respectively) for Creso (late cultivar) to −28.3% (−26.4% and 30.1% at Ussana and Benatzu, respectively) for Simeto (early cultivar) and −27.3% (−25.5% and −29.1% at Ussana and Benatzu, respectively) for Duilio (early cultivar) (Figure 4 and Figure 5). Comparing the average simulated grain yield results of the most drought prone cultivar (i.e., Creso) with the most resilient one (i.e., Duilio), a percentage gain of 3.9% in grain yield has been registered in favor of the latter. From this perspective, targeting cultivars onto different environments and climate conditions is one of the main adaptation strategies to climate change [22,94]. Furthermore, this study has another important implication: the cultivars considered for this study were released in Italy between thirty and forty years ago in different environmental conditions when compared to now. This means that: (i) a plethora of higher-yielding and better adapted cultivars is now available for current growing conditions; (ii) the importance of plant breeding in selecting superior genotypes ensuring good yield performances and yield stability in climate change conditions is paramount. Therefore, the role of cultivar choice in the short term and of plant breeding in the long term to tackle the detrimental effects of climate change on yield production and stability must be fully emphasized.

A last crucial issue is related to molecular responses in climate change conditions. Since global warming, heatwaves, droughts and a general decreasing rainfall trend are projected in the Mediterranean areas [1], identifying, sequencing and characterizing stress-inducible genes becomes essential to develop molecular markers for marker assisted breeding. Therefore, conventional breeding techniques and biotechnologies may increase the effectiveness of selection, allowing high-yielding and drought resistant genotypes. Focusing on a molecular approach, the results presented in this study confirm the important role of *DREB* genes in abiotic stress conditions as shown by Liu et al., in 1998 [65]. Furthermore, the importance of the dehydration-responsive factor gene (*TdDRF1*) has been confirmed both in greenhouse and in field conditions in other durum wheat and triticale cultivars [73]. Moreover, the link between grain yield, drought tolerance and specific polimorphism of the *TdDRF1* gene has been demonstrated in recent studies [95] and a correlation between grain yield and an increased expression of *TdDRF1.3* transcript in some drought tolerant and rustic durum wheat and triticale cultivars was found [96]. However, other studies must be addressed to explore the molecular mechanisms of regulation of the *TdDRF1* gene expression as well as the contribution of other genes.

## 5. Conclusions

This study highlights the importance of a multidisciplinary approach involving the use of crop modelling and biotechnology in order to predict and evaluate the performances of durum wheat genotypes under climate change conditions. Concerning crop modelling, CERES-Wheat proved to be an effective tool when used to predict grain yield, anthesis date and, to a lesser extent, kernel weight. The impact of climate change scenarios on grain yield is more negative, moving from mild to severe scenarios for all genotypes, but reductions are to some extent mitigated for Simeto and namely Duilio (early genotypes) in comparison with Creso (late genotype). On the other hand, kernel weight tends to increase slightly in response to increasing temperatures and decreasing rainfall in particular under mild climate change scenarios. All genotypes showed a reduction of their crop growing cycle as a consequence of increasing temperatures, with Duilio and Simeto revealing to be more resilient than Creso. The detrimental joint effect of simulated increasing temperatures and decreasing rainfall is also affected by soil fertility, with a stronger impact in low-yielding potential soils.

The predictive responses of the CERES-Wheat model can be also interpreted in the light of molecular responses of durum wheat cultivar to drought stress. In this context, the role of *DREB* genes, with a special focus on *TdDRF1*, in conditioning the resistance of genotypes to drought conditions must be underlined.

Furthermore, our analysis indicates that CERES-Wheat crop model responses are highly consistent with observations from most rain-fed durum wheat growing areas of the Mediterranean Region. By showing that early genotypes can be better adapted to increasing temperature and decreasing rainfall, the CERES-Wheat model proves to be a reliable tool to determine the impact of climate change on crops and can help to underline and quantify the simulated effects of cultivar choice to tackle downward trends in grain yield, particularly in the Mediterranean rain-fed areas. Hence, CERES-Wheat can be successfully used to support adaptation strategies such as targeting cultivars onto specific environments or to guide selection decisions in crop breeding programs, also implying the contribution of a molecular approach aiming at developing molecular markers for the resistance to abiotic and drought stresses.

## Figures and Tables

**Figure 1 genes-13-00488-f001:**
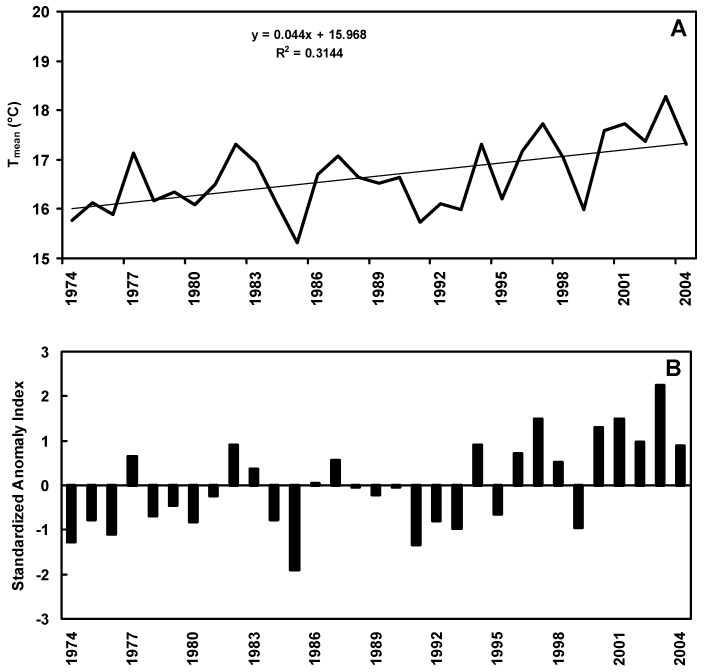
Trend of mean annual air temperatures (**A**) and Standardized Anomaly Index, SAI (**B**) over the period 1974–2004 at the AGRIS experimental station “S. Michele” (Southern Sardinia, Italy).

**Figure 2 genes-13-00488-f002:**
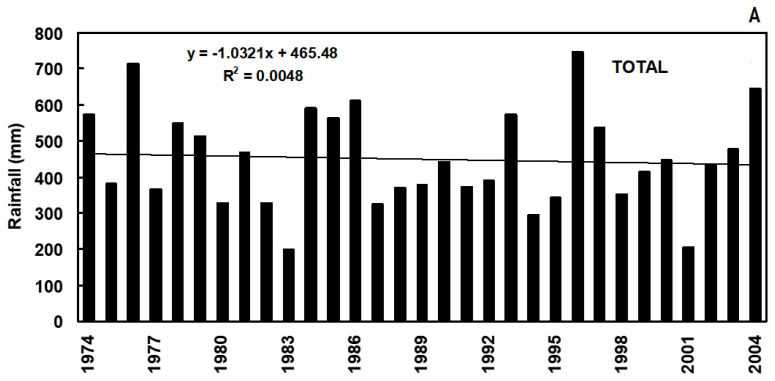
Annual (**A**) and seasonal (**B**) trends of rainfall over the period 1974–2004 at the AGRIS experimental station “S. Michele” (Southern Sardinia, Italy).

**Figure 3 genes-13-00488-f003:**
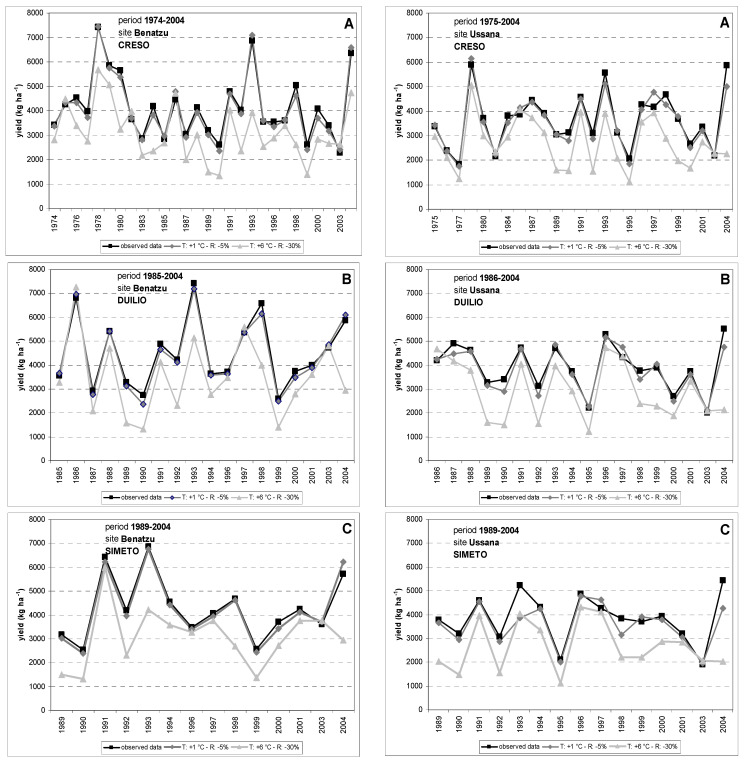
Effects of climate change scenarios on grain yield of durum wheat varieties Creso (**A**), Duilio (**B**), and Simeto (**C**) at Benatzu and Ussana sites. Simulation results from scenarios T1_R5 (temperature increase: +1 °C; rainfall reduction: 5%) and T6_R30 (temperature increase: +6 °C; rainfall reduction: 30%) are compared to observed yield data.

**Figure 4 genes-13-00488-f004:**
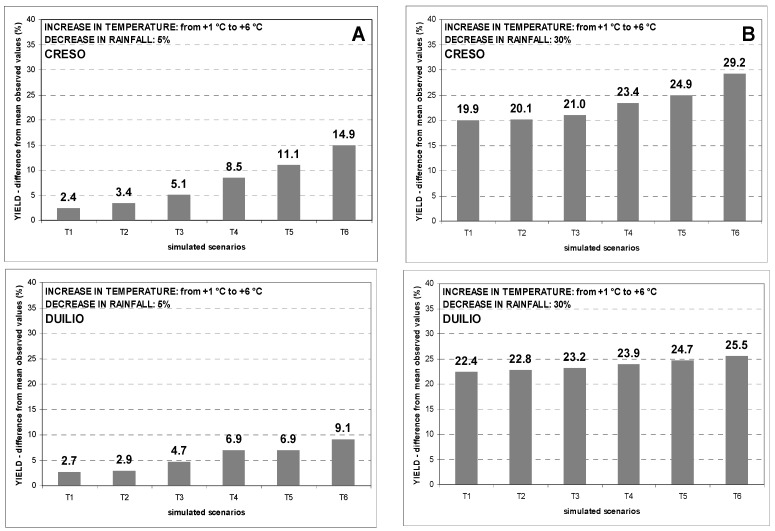
Percentage decline of grain yield over the period 1990–2004 at the experimental site of Benatzu for climate change scenarios characterized by increasing temperature (from +1 °C to +6 °C) and rainfall reduction by 5% (**A**) and 30% (**B**).

**Figure 5 genes-13-00488-f005:**
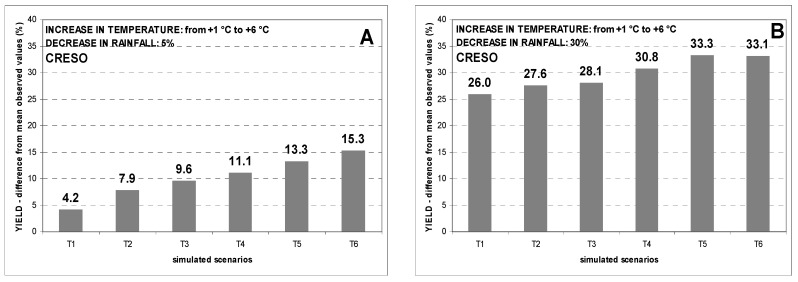
Percentage decline of grain yield over the period 1989–2004 at the experimental site of Ussana for climate change scenarios characterized by increasing temperature (from +1 °C to +6 °C) and rainfall reduction by 5% (**A**) and 30% (**B**).

**Figure 6 genes-13-00488-f006:**
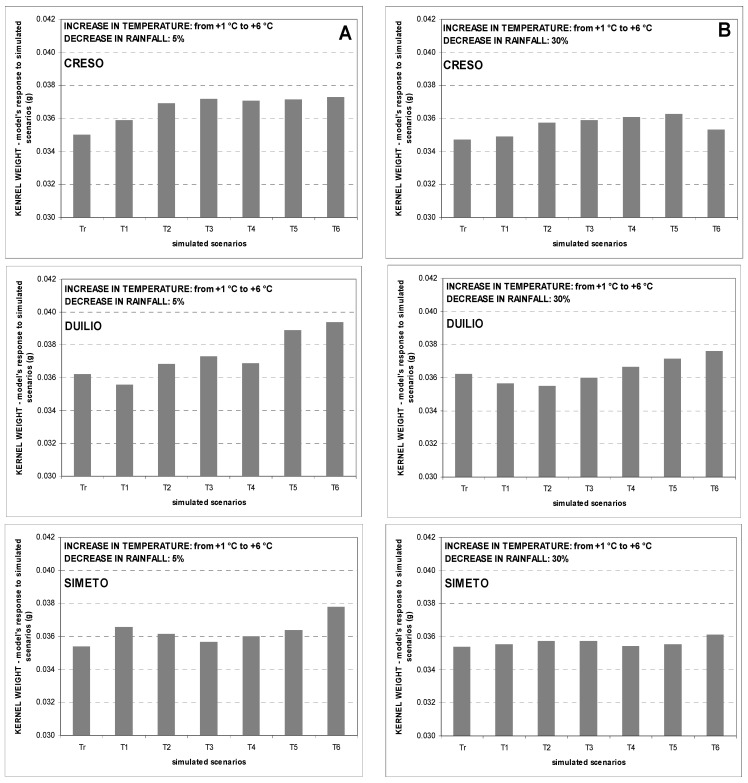
Average seed weight trends over the period 1990–2004 at the experimental site of Benatzu for climate change scenarios characterized by increasing temperature (from +1 °C to +6 °C) and rainfall reduction by 5% (**A**) and 30% (**B**). Tr = observed data.

**Figure 7 genes-13-00488-f007:**
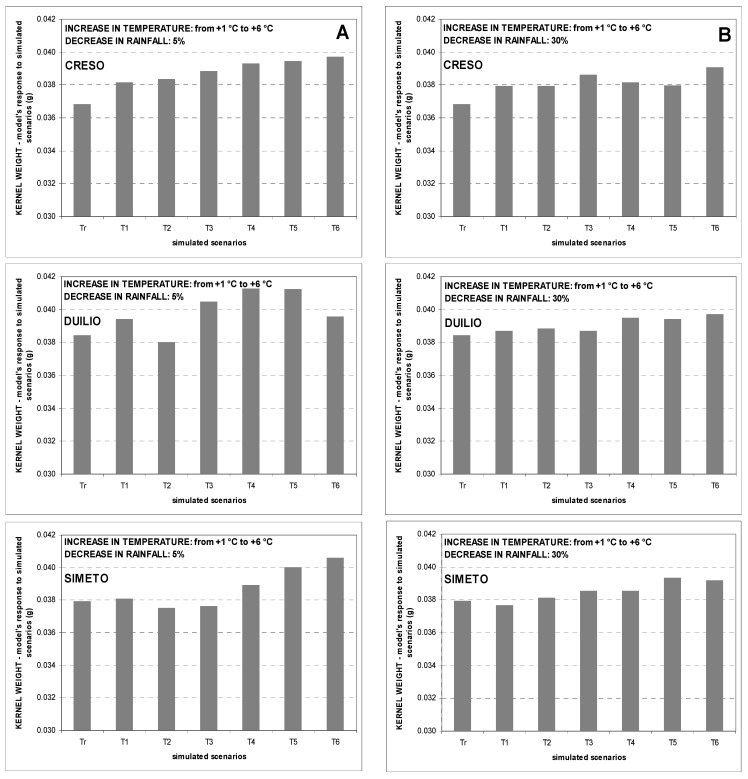
Average seed weight trends over the period 1989–2004 at the experimental site of Ussana for climate change scenarios characterized by increasing temperature (from +1 °C to +6 °C) and rainfall reduction by 5% (**A**) and 30% (**B**). Tr = observed data.

**Figure 8 genes-13-00488-f008:**
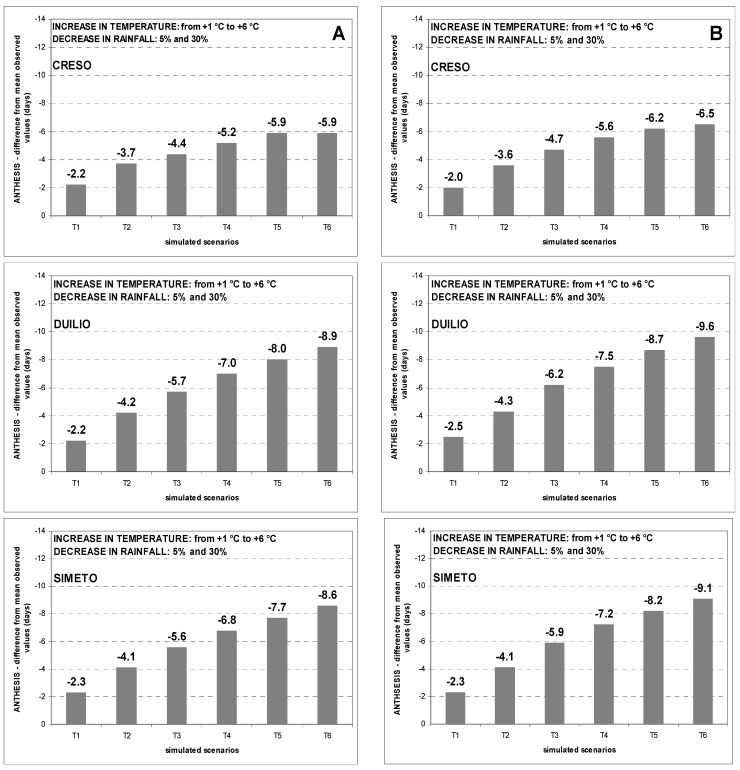
Reduction (in days) of crop growing cycle (from sowing to anthesis) over the period 1990–2004 at Benatzu (**A**) and the period 1989–2004 at Ussana (**B**) for climate change scenarios characterized by increasing temperature (from +1 °C to +6 °C) and rainfall reduction by 5% and 30%.

**Figure 9 genes-13-00488-f009:**
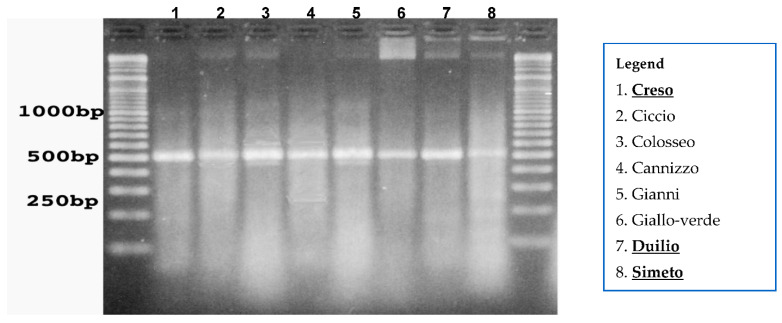
The expression pattern of RT-PCR with primer *drebfor1* e *drebrev1* of some durum wheat cultivars, including Creso (1), Duilio (7) and Simeto (8) analyzed at the 4th day without water.

**Table 1 genes-13-00488-t001:** Physical and chemical characteristics of soil at Benatzu and Ussana experimental sites.

	Benatzu	Ussana
Sand (%)	26.2	56.4
Silt (%)	34.4	21.5
Clay (%)	39.4	22.1
Texture	Clay Loam	Sandy Clay Loam
pH in H_2_O	8.5	7.9
Organic carbon (%)	1.62	0.83
Organic matter (%)	2.80	1.20
Total nitrogen (%)	0.15	0.07
C.E.C. (cmol kg^−1^)	2.9	2.3

**Table 2 genes-13-00488-t002:** Genetic coefficient values for Creso, Duilio, and Simeto durum wheat varieties during CERES-Wheat model calibration using data collected at two experimental sites located in Southern Sardinia, Italy [45]. P1D: Photoperiod sensitivity coefficient (% reduction/h near threshold); P1V: Vernalization sensitivity coefficient (%/d of unfulfilled vernalization); P5: Thermal time from the onset of linear filling to maturity (°C d); G1: Kernel number per unit stem + spike weight at anthesis (#/g); G2: Standard kernel size under optimum conditions (mg); G3: Standard, non-stressed dry weight (total, including grain) of a single tiller at maturity (g); PHINT: Thermal time between the appearance of leaf tips (°C d).

Genetic Coefficients	Creso	Duilio	Simeto
P1V	30.0	25.0	25.0
P1D	55.0	50.0	58.0
P5	450.0	480.0	450.0
G1	25.0	25.0	25.0
G2	55.0	55.0	55.0
G3	1.7	1.7	1.7
PHINT	100.0	90.0	90.0

**Table 3 genes-13-00488-t003:** Simulated climate change scenarios. R = rainfall; T = temperature.

	Decreasing Rainfall							
**Increasing Temperature**		**0**	**−5%**	**−10%**	**−15%**	**−20%**	**−25%**	**−30%**
	**0**	-	R5	R10	R15	R20	R25	R30
	**+1 °C**	T1	T1_R5	T1_R10	T1_R15	T1_R20	T1_R25	T1_R30
	**+2 °C**	T2	T2_R5	T2_R10	T2_R15	T2_R20	T2_R25	T2_R30
	**+3 °C**	T3	T3_R5	T3_R10	T3_R15	T3_R20	T3_R25	T3_R30
	**+4 °C**	T4	T4_R5	T4_R10	T4_R15	T4_R20	T4_R25	T4_R30
	**+5 °C**	T5	T5_R5	T5_R10	T5_R15	T5_R20	T5_R25	T5_R30
	**+6 °C**	T6	T6_R5	T6_R10	T6_R15	T6_R20	T6_R25	T6_R30

**Table 4 genes-13-00488-t004:** Simulated responses of three durum wheat cultivars (Creso—Cr, Duilio—Du, Simeto—Si) to two climate change scenarios (T1_R5 and T6_R30) at Benatzu (B) and Ussana (U) experimental sites in Sardinia, Italy. T1_R5 and T6_R30 scenarios project an average temperature increase of 1 °C and 6 °C, respectively, and an annual rainfall reduction of 5% and 30%, respectively. Simulation results (SIM) and means (M) of grain yield (kg ha^−1^), date of 2004 at Benatzu and 1989–2004 at Ussana.

		Grain Yield					Anthesis					Kernel Weight				
		OBS	SIM				OBS	SIM				OBS	SIM			
CV	Site		Scenario T1_R5		Scenario T6_R30			Scenario T1_R5		Scenario T6_R30			Scenario T1_R5		Scenario T6_R30	
		(kg ha^−1^)	(kg ha^−1^)	% Change	(kg ha^−1^)	% Change	(Dap)	(Dap)	Dap Change	(Dap)	Dap Change	(g)	(g)	% Change	(g)	% Change
Cr	B	4054	3955	−2.4	2869	−29.2	135	132	−3	127	−8	0.035	0.036	+2.5	0.035	0.0
	U	3700	3543	−4.2	2474	−33.1	141	138	−3	131	−10	0.037	0.038	+3.6	0.039	+6.0
	**M**	**3877**	**3749**	**−3.3**	**2672**	**−31.2**	**138**	**135**	**−3**	**129**	**−9**	**0.036**	**0.037**	**+3.0**	**0.037**	**+3.0**
Du	B	4573	4449	−2.7	3406	−25.5	129	127	−2	118	−11	0.036	0.036	0.0	0.038	+3.7
	U	3756	3633	−3.3	2662	−29.1	135	132	−3	122	−13	0.038	0.039	+2.6	0.040	+3.4
	**M**	**4165**	**4041**	**−3.0**	**3484**	**−27.3**	**132**	**130**	**−3**	**120**	**−12**	**0.037**	**0.038**	**+1.3**	**0.039**	**+3.6**
Si	B	4354	4280	−1.7	3206	−26.4	132	129	−3	121	−11	0.035	0.037	+3.3	0.036	+2.0
	U	3831	3575	−6.7	2676	−30.1	138	135	−3	125	−13	0.038	0.038	0.0	0.039	+3.2
	**M**	**4093**	**3928**	**−4.2**	**2941**	**−28.3**	**135**	**132**	**−3**	**123**	**−12**	**0.037**	**0.038**	**+1.7**	**0.038**	**+2.6**
Mean	B	4327	4228	−2.3	3460	−27.0	132	129	−2.7	122	−10	0.035	0.036	+1.9	0.036	+1.9
	U	3762	3584	−4.7	2604	−30.8	138	135	−3.0	126	−12	0.038	0.038	+2.1	0.039	+4.2
	**M**	**4045**	**3906**	**−3.5**	**3032**	**−28.9**	**135**	**132**	**−2.9**	**124**	**−11**	**0.037**	**0.037**	**+2.0**	**0.038**	**+3.1**

Legend: OBS—Observed data; SIM—Simulated data, CV—Cultivar, Cr—Creso, Du—Duilio, Si—Simeto.

## Data Availability

The data presented in this study are available on request from the corresponding author.

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
