# Peer review of "Performance Prediction of Durum Wheat Genotypes in Response to Drought and Heat in Climate Change Conditions"

_genes, 2022, doi:10.3390/genes13030488_

Round 1

Reviewer 1 Report

This manuscript reports the results of the monitoring and analysis of the field performance of 3 elite Italian durum wheat varieties in combination with the climatic data registered on the location (South Sardinia, IT). The CERES-Wheat model tool, which specifically allows to simulate wheat growth in response to climate, soil, genotypes and management, has been used here to simulate the effects of increasing temperature and decreasing rainfall on grain yield, grain size and anthesis date. One main strength of this work is the long period of data availability, almost 30 years from 1973 to 2004. Interestingly, the discussion of the agronomic performance in the different (48) climatic scenarios has been integrated and enriched with molecular data previously available on the plant molecular response to dehydration based on specific DREB genes, known to be related to water stress and other abiotic threats.

In my opinion, the manuscript is endowed of an interesting and well-told content, presented in an accurate way. In general, I found the reading of this work very nice and smooth. The reported results are rightly supported from the data collected and analyzed. Finally, the manuscript deserves to be published, it needs only very few improvements. Just a few suggestions are listed below.

In the ABSTRACT

- Line 22: For clarity, you could mention here (or above in line 13) the names of the two experimental sites, Benatzu and Ussana.

In MATERIALS AND METHODS

- Line 168: could you please explicit what is the data from which the global radiation has been estimated by RadEst?

- Concerning Table 1, the methods used for the soil analyses have not been reported, but this information is generally useful for the readers. If the analyses have been performed by an external company, could you please mention it in the text (or in the legend of Table 1), so that it could be possible for the readers to search/find this information? Otherwise, you are suggested to report - briefly - the different procedures that have been applied. Moreover, you could delete the line related to the % stones, since it is 0%. I also suggest, if the data is available, to report the organic matter % with two decimal digits after the comma. Lastly, there is no need to report the acronym for C.E.C. also in the legend of the table if it has already been mentioned in the text.

Fig. 1 is not very informative, maybe it can be eliminated.

Author Response

The authors are grateful to the Reviewer whose suggestions helped us considerably to improve the manuscript.

Please see the attachments for our point-by-point responses.

Sincerely,

Marco Dettori

Reviewer 2 Report

The manuscript assessed agronomic traits of three durum cultivars under two environments in years 1973-2004, which validated the CERES model and the important role of DREB genes.  Specific comments are as followed:

  1. Please be more specific about “abiotic stress” in the Title, which is too general. This manuscript focused only on drought stress and heat stress.
  2. The paragraphs in Discussion (also in Materials and Methods) need to be reorganized --- Many paragraphs only have one sentence. For example, Page 24 “Model accuracy in…”, Page 25 “ the Yield reductions…”, Page 6 “Table 2 shows the …”.
  3. Page 5 lines 191 “DREB genes”: please indicate/list how many DREB genes (and their gene names --- better use IWGSC gene names) you have used/referred in this study.
  4. Page 6, please delete the sentence “Table 2 shows…”. Instead, please cite Table in the text.
  5. Please combine Figure 4 & 5. I feel Figure 6 & 7, and Figure 8 & 9 can also be combined if the legends/descriptions in the figures can be more concise.
  6. Page 23 line 531 “showed an intense band at 500 bp”: Is this band from one gene or from several genes? What are the gene names? Is there a figure/picture to show this result?
  7. Page 24 lines 550-551 “hinder the timing of sowing especially when associated with increasing mean precipitation per wet day”: if this sentence means that heavy rain hinders sowing, then please use simpler sentence to say it.
  8. Page 28 line 626 “TdDRF1”: Please give the IWGSC name of this gene. Please indicate why it is important. For example, how much expression level difference it shows under control and drought? Why this gene is more outstanding than other DREB genes?

Author Response

The authors are grateful to the Reviewer whose suggestions helped us considerably to improve the manuscript.

Please see the attachment for our point-by-point response.

Sincerely,

Marco Dettori
